# Estimating health spending associated with chronic multimorbidity in 2018: An observational study among adults in the United States

**Angela Y. Chang** [1,2,3]*, **Dana Bryazka** [4], **Joseph L. Dieleman** [4]

1 Danish Institute for Advanced Study, University of Southern Denmark, Copenhagen, Denmark,
2 Department of Clinical Research, University of Southern Denmark, Copenhagen, Denmark,
3 Interdisciplinary Centre on Population Dynamics, University of Southern Denmark, Odense, Denmark,
4 Institute for Health Metrics and Evaluation, University of Washington, Seattle, Washington, United States of America

* achang@health.sdu.dk

## Abstract

**Data Availability Statement:** Data used in this paper cannot be made available due to contractual restrictions but can be purchased from IBM MarketScan Research Database. It provides claim-level health care information on millions of commercially insured enrollees below age 65.

### Background

The rise in health spending in the United States and the prevalence of multimorbidity—having more than one chronic condition—are interlinked but not well understood. Multimorbidity is believed to have an impact on an individual's health spending, but how having one specific additional condition impacts spending is not well established. Moreover, most studies estimating spending for single diseases rarely adjust for multimorbidity. Having more accurate estimates of spending associated with each disease and different combinations could aid policymakers in designing prevention policies to more effectively reduce national health spending. This study explores the relationship between multimorbidity and spending from two distinct perspectives: (1) quantifying spending on different disease combinations; and (2) assessing how spending on a single diseases changes when we consider the contribution of multimorbidity (i.e., additional/reduced spending that could be attributed in the presence of other chronic conditions).

### Methods and findings

We used data on private claims from Truven Health MarketScan Research Database, with 16,288,894 unique enrollees ages 18 to 64 from the US, and their annual inpatient and outpatient diagnoses and spending from 2018. We selected conditions that have an average duration of greater than one year among all Global Burden of Disease causes. We used penalized linear regression with stochastic gradient descent approach to assess relationship between spending and multimorbidity, including all possible disease combinations with two or three different conditions (dyads and triads) and for each condition after multimorbidity adjustment. We decomposed the change in multimorbidity-adjusted spending by the type of combination (single, dyads, and triads) and multimorbidity disease category.

Contact can be made through this website: https://www.merative.com/contact.

**Funding:** Research reported in this publication was supported by the National Institute on Aging of the National Institutes of Health (Award Number P30AG047845 to JLD and AYC). The funders had no role in study design, data collection and analysis, decision to publish, or preparation of the manuscript.

**Competing interests:** The authors have declared that no competing interests exist.

**Abbreviations:** CMS, Centers for Medicare and Medicaid Services; COPD, chronic obstructive pulmonary disease; DEX, Disease Expenditure Project; EMBI, endocrine, metabolic; blood, and immune; GBD, Global Burden of Disease; ICD-10, International Classification of Diseases, 10th Revision; IHD, ischemic heart disease; IHME, Institute for Health Metrics and Evaluation; SGD, stochastic gradient descent; UI, uncertainty interval.

We defined 63 chronic conditions and observed that 56.2% of the study population had at least two chronic conditions. Approximately 60.1% of disease combinations had super-additive spending (e.g., spending for the combination was significantly greater than the sum of the individual diseases), 15.7% had additive spending, and 23.6% had sub-additive spending (e.g., spending for the combination was significantly less than the sum of the individual diseases). Relatively frequent disease combinations (higher observed prevalence) with high estimated spending were combinations that included endocrine, metabolic, blood, and immune disorders (EMBI disorders), chronic kidney disease, anemias, and blood cancers. When looking at multimorbidity-adjusted spending for single diseases, the following had the highest spending per treated patient and were among those with high observed prevalence: chronic kidney disease ($14,376 [12,291,16,670]), cirrhosis ($6,465 [6,090,6,930]), ischemic heart disease (IHD)-related heart conditions ($6,029 [5,529,6,529]), and inflammatory bowel disease ($4,697 [4,594,4,813]). Relative to unadjusted single-disease spending estimates, 50 conditions had higher spending after adjusting for multimorbidity, 7 had less than 5% difference, and 6 had lower spending after adjustment.

## Conclusions

We consistently found chronic kidney disease and IHD to be associated with high spending per treated case, high observed prevalence, and contributing the most to spending when in combination with other chronic conditions. In the midst of a surging health spending globally, and especially in the US, pinpointing high-prevalence, high-spending conditions and disease combinations, as especially conditions that are associated with larger super-additive spending, could help policymakers, insurers, and providers prioritize and design interventions to improve treatment effectiveness and reduce spending.

## Author summary

### Why was this study done?

- Many would agree that much health spending is directed towards complex cases that include a combination of multiple chronic conditions, but existing literature estimating disease-specific spending generally fail to systematically account for multimorbidity.

- Few studies have explored whether different combinations of conditions lead to greater or less spending than the sum of having the diseases separately.

### What did the researchers do and find?

- We used a large claims dataset of over 16 million commercially insurance US working population in 2018 to study the relationship between annual health spending and multimorbidity.

- We developed a novel approach to adjust spending for each disease for multimorbidity (i.e., estimating the additional/reduced spending that could be attributed in the presence

of other conditions) and found that most diseases have higher estimated spending after adjustment.

- We further found that chronic kidney disease, ischemic heart disease-related heart conditions, cirrhosis, and inflammatory bowel disease are associated with high spending per treated case, high observed prevalence, and contribute the most to spending when in combination with other chronic conditions.

**What do these findings mean?**

- Multimorbidity adjustments should be performed for any health spending analysis, otherwise researchers will likely largely underestimate spending for most diseases while overestimating for the remaining diseases.

- In the midst of a surging health spending globally, and especially in the United States, pinpointing high-prevalence, high-spending conditions and super-additive disease combinations could help policymakers design interventions to improve treatment effectiveness and reduce spending.

## Introduction

There are many concerning trends in healthcare in the United States. One is the magnitude and rapid growth of health spending, estimated at nearly 20% of the US economy in 2020, which has more than doubled in the past two decades [1]. Another is the rise in the burden of multimorbidity, commonly defined as the coexistence of two or more chronic conditions [2,3]. These two trends are likely interrelated, yet there is lack of good understanding of the relationship between health spending and multimorbidity. For example, studies have found that having certain conditions may affect the treatment of other comorbidities, and it is possible that the disease combinations would lead to a higher or lower spending beyond the sum of the spending of single conditions [4,5]. Without considering the impact of multimorbidity on spending, we are likely missing the opportunity for synergistically and efficiently tackling these issues [6].

Better insights into how and why health spending is rapidly increasing could in turn help slow down its growth. With the rise in multimorbidity, however, it has become more difficult to accurately associate spending to single conditions. Upon reviewing previous literature of similar inquiries and recent systematic reviews on the cost of multimorbidity, we identified four main gaps in estimating spending in the context of multimorbidity [7,8]. First, most studies apply a simple definition of multimorbidity, commonly the count of conditions an individual has in addition to a base condition, and do not consider which additional diseases are being bundled [8–13]. For example, one study found that one additional chronic condition was associated with nearly double the annual health spending [14]. However, this information is not enough for policymaking. It is reasonable to hypothesize that a person with diabetes and hypertension would incur different levels of spending than another person with diabetes and depression, yet the relationship between spending and the types of disease combinations are rarely explored. Second, most studies only include a small set of chronic conditions, based on

convenience sample or high prevalence [3,12,15,16]. This limitation is mostly due to data availability (data sources not reporting more conditions) or computational restrictions (inadequate methods or insufficient computational power to analyze larger sets of disease combinations) [17]. Third, studies like those published by the Centers for Medicare and Medicaid Services (CMS) estimated per capita spending for dyads and triads of 20 chronic conditions, but they did not estimate spending for each disease after adjusting for multimorbidity (i.e., how spending for combinations can be attributed back to single conditions) [18]. This was also presented in a recent systematic review and meta-analysis, in which the authors were able to calculate the mean cost data of only 11 common dyads, and none of the reviewed studies reported whether the combinations have additive, sub-, or super-additive spending [7]. Finally, some studies focus on a single condition and estimate effects of comorbidities on spending of this particular condition, but it is difficult to combine the results of these studies to compare across diseases due to vastly different study designs [11,19–21]. The most similar study design is by Dieleman and colleagues, which includes a comprehensive set of conditions and estimate spending after reallocating resources based for comorbidities, but it still retains focus on individual health conditions rather than the spending associated with multimorbidity [22].

This study attempts to fill these literature gaps and propose a novel approach to understanding this topic. First, instead of merely counting the number of chronic conditions an individual has, we are interested in how different types of multimorbidity combinations—for example, cardiovascular diseases + mental disorders or dementia + musculoskeletal disorders—lead to different spending outcomes. Previous studies found that multimorbidity leads to higher spending, but whether having one additional chronic condition (and the type of additional condition) leads to a super-additive rather than simply additive effect on health spending is unclear. Furthermore, to our knowledge, no study in the US has shown a synergistic effect between chronic conditions—where health spending among adults with two conditions is less than the simple addition of spending associated with the single conditions. We also estimated spending associated with single chronic conditions by considering the contribution of multimorbidity, i.e., additional spending that could be attributed in the presence of other chronic conditions. From here onwards, we call this the "multimorbidity-adjusted spending." Second, we used the comprehensive list of conditions from the Global Burden of Disease (GBD) 2019 study to create all possible two- and three-way combinations [23]. The use of a comprehensive list of chronic conditions gave us a more accurate set of spending estimates that is also comparable across conditions.

## Methods

This study is reported following the REporting of studies Conducted using Observational Routinely collected health Data (RECORD) guideline.

### Data source

The Truven Health MarketScan Commercial Claims and Encounters Database provides claim-level healthcare information on millions of commercially insured enrollees below age 65 [24,25]. We used claims data from the 2018 inpatient services, outpatient services, and inpatient admissions tables. We restricted the study sample to adults between ages 18 and 64 with unique enrollee identification numbers. To our knowledge, the differences in patients' sociodemographic characteristics and the universe of all privately insured individuals has not been assessed. For example, Truven stated that the data mostly come from large employers, and thus individuals employed in medium and small firms may be underrepresented [26]. We excluded enrollees with mismatching demographic data, such as people who had two birth

years or were assigned both male and female in different claim records. We further excluded enrollees whose spending data were missing or were negative; enrollees with claim records that had zero spending were not excluded.

### Defining and assigning chronic condition diagnoses

The GBD 2019 cause hierarchy includes 297 most-detailed diseases and injuries [23]. First, to identify chronic conditions, we selected with an average duration of greater than one year, and injuries were excluded from this analysis [27]. Second, to allow mapping between the database and GBD causes, we collected all International Classification of Diseases, 10th Revision, Clinical Modification codes (ICD-10) associated with each chronic cause previously conducted by the Institute for Health Metrics and Evaluation (IHME) US Disease Expenditure Project (DEX) [1]. Third, to improve data efficiency by reducing the number of covariates in the regression model, we combined a subset of more-detailed conditions with lower observed prevalence into larger disease categories (see Table A2 in S1 Appendix). For example, all alcohol and drug use-related mental disorders (such as opioid, cocaine, amphetamine, cannabis, and other drug use) were grouped into one; esophageal and stomach cancers were combined and renamed as upper gastrointestinal cancers. Fourth, we applied the algorithm set by the CMS Chronic Conditions Data Warehouse, which qualifies an ICD code to be associated with a chronic condition if it is present in at least one inpatient or two outpatient claims [28,29]. Finally, we ran through all claims and considered an individual to have the chronic condition if the criteria described above were met. We included all diagnoses code associated with all claims records and did not restrict the analysis to only primary diagnosis or a small subset of diagnoses codes.

For comprehensiveness, GBD and DEX have assigned residual "other" categories, such as "other chronic respiratory diseases" and "other neoplasms," although these categories are generally poorly defined. In this study, we included these "other" conditions in the statistical analysis but did not present the results in the main paper. The full results, including these "other" categories, are reported in Table A5 in S1 Appendix.

### Estimating annual spending per enrollee

We estimated annual insurance spending for each enrollee by adding all net payments reported in 2018. The net payment for each claim, as defined by the data source, is the payment to a provider for a service, calculated by removing deductibles, coinsurance, and coordination of benefits and other savings from gross covered payment [24]. We included spending on all claims assigned to the person, regardless of whether the claim was associated with a chronic condition. Including both chronic and non-chronic spending is necessary to be able to capture the potential effects of having chronic conditions on the individual's overall health outcomes, health seeking behaviors, and ultimately, health spending. Spending was transformed on a natural logarithmic scale. All estimates are presented in 2018 US dollars.

Estimating super-additive, additive, and sub-additive effects of chronic conditions on annual health spending

This study took a person-based regression approach—regressing a person's total 2018 health spending on health conditions indicators—in estimating spending per treated case [30,31]. We applied the following linear regression model:

$$spend_i = \beta_{0i} + \sum_{j=1}^{J} \beta_{ij} dx_{ij} + \sum_{k=1}^{K} \beta_{ik} \, dyad_{ik} + \sum_{l=1}^{L} \beta_{il} triad_{il} + age_i + sex_i$$
$$+ \, region \; dummies_i + \varepsilon_i \qquad (1)$$

where $i$ is enrollee, $dx_j$ is each chronic condition, $dyad_k$ and $triad_l$ represent the interaction terms for all possible chronic condition dyads and triads, respectively, $age_i$, $sex_i$, and $region_i$ represent the enrollee's age group (10-year age groups), sex, region of residence (Northeast, North Central, South, West, and Unknown), and $\varepsilon_i$ is the error term [14]. This equation estimates the spending associated with single conditions and different combinations. For example, an estimated positive and statistically significant $\hat{\beta}_{ik}$ suggests that the combination has a super-additive effect on spending, which is greater than the sum of the spending associated with having these conditions separately. If $\hat{\beta}_{ik}$ is not statistically different from zero, it would suggest that the combination has an additive effect, and a statistically significant negative coefficient suggests that the combination has a synergistic, negative effect, meaning that spending is less than the sum of the spending associated with having the conditions separately.

Given the large size of the dataset (6.7+ billion claims), over 40,000 covariates representing all possible dyads and triads of chronic conditions, and the need for strong computational power, we applied a regression framework using the stochastic gradient descent (SGD) approach, a commonly used method in solving large machine learning problems. SGD updates the regression coefficients iteratively to minimize the objective function for the regression model of interest (minimize mean squared error), using a smaller batch of data for each iteration. The general concept and objective of this approach is close to that of a typical ordinary least squares regressions [32]. To prevent having too many variables in the model that have small contributions, we applied a lasso penalized regression model to shrink the coefficient values of these covariates. To ensure stability of the model results, we conducted 50 SGD model runs and bootstrapped the results across runs for 10,000 times to get the estimates for all coefficients. Details on the model and parameter setting can be found in Section 3A of S1 Appendix.

Estimates of total spending associated with any combination were derived by adding the coefficients for the single conditions independently and the coefficients from the interaction terms (from the combinations). For example, total spending for diabetes + osteoarthritis was calculated as the sum of the coefficient for diabetes, coefficient for osteoarthritis, and coefficient for the interaction term diabetes * osteoarthritis. For triads, we further added the three dyads to the sum.

Estimating spending associated with each individual health condition, adjusting for multimorbidity ("multimorbidity-adjusted spending")

For the second outcome of interest, we are interested in estimating the proportion of spending for the combination that could be attributed back to single conditions. For example, we would have a more accurate spending on diabetes because we would have not only the diabetes-specific spending but also the additional or reduced amount of spending diabetes incurs when in combination with other conditions. More specifically, the coefficient of the interaction term for the diabetes–osteoarthritis combination needs to be split into one part associated with diabetes and another with osteoarthritis. A four-step process was implemented to do so:

First, we ran a linear regression model among study population with diabetes to estimate the effect of having osteoarthritis on annual health spending:

$$spend_j = \beta_{0j} + \sum_{k=1}^{K-1} \beta_{jk} dx_{jk} + age_j + sex_j + region_j + \varepsilon_j \qquad (2)$$

where $j$ is the enrollee with the disease (diabetes in this example), and $dx_k$ is each additional chronic condition beyond diabetes. With this equation, we derive $\beta_{osteoarthritis|diabetes}$, the coefficient representing the effect of having osteoarthritis on spending among people with diabetes. We ran the same model for people with osteoarthritis to derive $\beta_{diabetes|osteoarthritis}$, the coefficient representing the effect of having diabetes on spending among people with osteoarthritis.

Second, we took the coefficient of the interaction term for diabetes and osteoarthritis, $\beta_{diabetes,osteoarthritis}$ (derived from Eq 1), and split the coefficient into two parts:

$$\beta_{diabetes,osteoarthritis} = \beta_{diabetes|combination} + \beta_{osteoarthritis|combination} \tag{3}$$

$$= \beta_{diabetes,osteoarthritis} \times \frac{\beta_{diabetes|osteoarthritis}}{\beta_{diabetes|osteoarthritis} + \beta_{osteoarthritis|diabetes}}$$

$$+ \beta_{diabetes,osteoarthritis} \times \frac{\beta_{osteoarthritis|diabetes}}{\beta_{diabetes|osteoarthritis} + \beta_{osteoarthritis|diabetes}}$$

where $\beta_{diabetes|combination}$ is the estimated part of the interaction coefficient that is attributed to diabetes, and $\beta_{osteoarthritis|combination}$ is the part attributed to osteoarthritis. $\beta_{diabetes|osteoarthritis}$ is the coefficient from Eq (2) on the indicator variable of whether those with osteoarthritis also have diabetes as a comorbidity, and $\beta_{osteoarthritis|diabetes}$ is the coefficient on the indicator variable of whether those with diabetes also have osteoarthritis as a comorbidity. To estimate spending among all interaction coefficients that should be attributed to having diabetes, we repeated the previous steps for all conditions that co-occurred with diabetes.

Third, we calculated prevalence, i.e., the probability of the disease combinations occurring among people with diabetes (for example, the probability of someone with diabetes also having osteoarthritis). This is needed for the next step, in which we adjusted each spending estimates based on its prevalence, such that more common combinations of diabetes and another disease receives a higher weight than combinations that are less common. This was done by multiplying the spending associated with the combination with its prevalence from step 3 and summed across disease combinations. This final figure is the part of the spending associated with dyads that should be attributed to diabetes. The same approach was extended to combinations of three, in which we ran the model among people with two conditions and took the coefficient on the indicator variable of having the third condition (explained in more detail in Section 2 in S1 Appendix).

For the purpose of comparison, we estimated the non-adjusted spending for each condition by using the same regression model in Eq (1) but without the disease interaction terms ($\sum_{k=1}^{K} \beta_{ik} dyad_{ik}$ and $\sum_{l=1}^{L} \beta_{il} triad_{il}$). The results from this simple model (referred in the results as "non-adjusted spending") was then used to compare against the main results (multimorbidity-adjusted spending).

We decomposed the change in multimorbidity-adjusted spending by the type of combination (single, dyads, and triads) and the multimorbidity disease categories (e.g., cardiovascular, neoplasms).

For the purpose of reporting, we present estimates of health spending for a 35- to 44-year-old female from the South region, which reflects the most common age, sex, and regional characteristics of the study population. We also report the coefficients for all demographic covariates in Table A4 in S1 Appendix. For estimates for disease combinations, only those with observed prevalence greater than 50 per 100,000 people are listed in the figures and tables.

## Quantifying uncertainty

First, to generate 95% uncertainty interval (UI) for spending associated with disease combinations, we bootstrapped the means from all the model runs for 10,000 times. Second, to generate UI for multimorbidity-adjusted spending for each single condition, we ran Monte Carlo simulations ($n$ = 1,000 draws) while varying the estimates associated with the combination and the proportion of combination attributed to each single condition.

Analyses were performed using R, version 4.0.5 (R Foundation for Statistical Computing, Vienna, Austria), and Python 3.8.1 (Python Software Foundation, Hampton, New Hampshire, USA).

## Results

### Selected chronic conditions

A total of 166 most-detailed GBD causes were determined as chronic (listed in Table A1 in S1 Appendix), of which we reduced down to 63 by combining them into larger disease categories to improve data efficiency (Table 1).

### Study population

A total of 16,288,894 enrollees and their 6,726,532,451 claims were included in the analysis. Population characteristics are presented in Table 2: 56.2% were female, mean age was 42.3 years, the number of chronic conditions for an individual ranged from 0 to 32, with a mean of 2.6 conditions. Approximately 23.6% enrollees had no chronic conditions, 20.2% had one condition, and the remaining 56.2% had two or more chronic conditions, of which 94.0% of them had more than three conditions. Looking at single chronic conditions, skin and subcutaneous diseases (21,662 per 100,000; 21.7% of all study population), hypertension (17,727; 17.7%), gynecological diseases (15,033, 15.0%), musculoskeletal pain (14,953, 15.0%), and hyperlipidemia (13,465; 13.5%) had the highest observed prevalence rates. Looking at combinations of chronic conditions, the most common health condition dyads were hyperlipidemia + hypertension (7,177; 7.2%), diabetes + hypertension (5,044; 5.0%), hypertension + skin and subcutaneous diseases (4739; 4.7%), diabetes + hyperlipidemia (4,625; 4.6%), and gynecological diseases + skin and subcutaneous diseases (4,589; 4.6%); the most frequent health condition triads included two high-prevalence risk factors (hyperlipidemia + hypertension) plus one of the following chronic conditions: diabetes (3,069; 3.1%), skin and subcutaneous diseases (2,065; 2.1%), obesity (1,914; 1.9%), endocrine, metabolic, blood, and immune disorders (EMBI disorders) (1,888; 1.9%), and diabetes + hypertension + obesity (1,571; 1.6%). The mean and median annual health spending of the study population were $6,388 and $633, respectively.

### Spending for combinations of chronic conditions

Out of 41,664 possible health condition combinations (dyads or triads), regression coefficients for 25,277 (60.1%) were positive (super-additive), 6,553 (15.7%) were nearly zero (additive; between −1 and 1), and 9,834 (23.6%) were negative (sub-additive). Among health condition combinations with observed prevalence rate greater than 50 per 100,000, the five largest super-additive spending were found in combinations of blood cancers + hemoglobinopathies and hemolytic anemias (henceforth anemias) (+$3,227, 95% UI [2,541,3,905]), chronic kidney disease + EMBI disorders + anemias (+$3,111 [2,679,3,535]), chronic kidney disease + anemias (+$3,074 [2,718,3,431]), blood cancers + EMBI disorders (+$3,017 [2,427,3,591]), chronic kidney disease + EMBI disorders (+$2,887 [2,617,3,148]). The five largest sub-additive spending were found in EMBI disorders + hyperlipidemia (−$733 [−851,−620]), hyperlipidemia + anemias (−$702 [−885,−516]), cirrhosis + hyperlipidemia (−$610 [−838,−370]), hyperlipidemia + anemias + skin and subcutaneous diseases (−$558 [−692,−416]), and chronic kidney disease + low back and neck pain + hyperlipidemia (−$545 [−727,−363]). More details are provided in Table A6 in S1 Appendix.

**Table 1. Observed prevalence, proportion of multimorbidity, multimorbidity-adjusted annual per capita spending, and the comorbidities with the highest attributable spending for all chronic conditions.**

| Chronic condition | Observed prevalence rate (per 100,000) | Proportion with additional chronic condition | Multimorbidity-adjusted annual spending per treated case | Ratio of multimorbidity-adjusted spending and non-adjusted spending |
|---|---|---|---|---|
| *Neoplasms* | | | | |
| Bladder and kidney cancers | 173.7 | 95.1% | $1,536 [1,421–1,672] | 1.6 |
| Blood cancers | 218.7 | 94.8% | $9,387 [8,627–10,071] | 2.8 |
| Brain and nervous system cancer | 40.2 | 95.9% | $6,298 [5,334–7,386] | 3.7 |
| Breast cancer | 5,169.2 | 92.2% | $579 [516–655] | 0.6 |
| Colon and rectum cancer | 3,145.2 | 95.3% | $1,132 [949–1,285] | 1.4 |
| Ear, nose, throat cancers | 32.7 | 97.2% | $1,406 [1,223–1,567] | 2.4 |
| Reproductive organ cancers | 3,945.4 | 87.9% | $837 [723–960] | 1.8 |
| Gastrointestinal gland cancers | 41.4 | 98.6% | $3,462 [3,078–3,904] | 2.6 |
| Hodgkin lymphoma | 34.9 | 91.4% | $1,191 [1,086–1,316 | 1.8 |
| Lip and oral cavity cancers | 31.1 | 97.0% | $1,536 [1,374–1,680] | 2.3 |
| Skin cancers | 790.5 | 98.2% | $806 [644–946] | 1.2 |
| Thyroid cancer | 202.0 | 93.7% | $534 [460–592] | 1.2 |
| Tracheal, bronchus, and lung cancer | 94.6 | 98.4% | $5,422 [4,874–6,085] | 2.9 |
| Upper gastrointestinal cancers | 21.2 | 97.7% | $1,917 [1,788–2,056] | 2.2 |
| *Cardiovascular diseases* | | | | |
| Atrial fibrillation and flutter | 642.2 | 96.9% | $5,147 [4,864–5,475] | 3.1 |
| IHD-related heart conditions | 2,228.2 | 97.7% | $6,029 [5,529–6,529] | 1.9 |
| Peripheral vascular disease | 457.8 | 98.4% | $1,265 [1,175–1,338] | 1.3 |
| Rheumatic heart disease | 137.1 | 98.6% | $4,268 [4,17–4,472] | 2.9 |
| Stroke | 698.8 | 98.7% | $5,395 [4,564–6,326] | 2.3 |
| *Chronic respiratory diseases* | | | | |
| Asthma | 3,020.4 | 93.3% | $1,277 [1,238–1,323] | 0.9 |
| COPD | 723.3 | 97.8% | $1,496 [1,399–1,601] | 1.0 |
| Interstitial lung disease and pulmonary sarcoidosis | 170.2 | 97.7% | $1,024 [950–1,113] | 0.8 |
| *Digestive diseases* | | | | |
| Cirrhosis | 1,536.8 | 96.7% | $6,465 [6,090–6,930] | 2.3 |
| Gallbladder and biliary diseases | 801.8 | 93.7% | $7,509 [6,825–8,124] | 2.3 |
| Gastritis and duodenitis, peptic ulcer disease | 5,890.0 | 95.4% | $2,550 [2,484–2,612] | 1.4 |
| Inflammatory bowel disease | 1,030.6 | 89.0% | $4,697 [4,594–4,813] | 1.8 |
| Inguinal, femoral, and abdominal hernia | 977.5 | 93.6% | $2,608 [2,608–2,608] | 1.2 |
| *Neurological disorders* | | | | |
| Alzheimer's disease and other dementias | 106.6 | 98.0% | $896 [785–1,19] | 1.2 |
| Epilepsy | 430.7 | 89.5% | $1,882 [1,765–2,19] | 1.1 |
| Headache disorders | 2,797.6 | 94.6% | $563 [489–624] | 0.7 |
| Multiple sclerosis | 226.9 | 91.9% | $1,532 [1,403–1,669] | 1.1 |
| Parkinson's disease | 49.4 | 94.8% | $307 [257–364] | 1.1 |
| *Mental disorders* | | | | |
| Anxiety disorders | 10,574.6 | 89.9% | $1,087 [996–1,163] | 3.0 |
| Attention-deficit/hyperactivity disorder | 2,059.2 | 84.9% | $137 [127–145] | 1.1 |
| Bipolar disorder | 779.5 | 93.2% | $1,008 [914–1,087] | 1.6 |

(*Continued*)

**Table 1.** (Continued)

| Chronic condition | Observed prevalence rate (per 100,000) | Proportion with additional chronic condition | Multimorbidity-adjusted annual spending per treated case | Ratio of multimorbidity-adjusted spending and non-adjusted spending |
|---|---|---|---|---|
| Depressive disorders | 5,853.0 | 94.0% | $963 [804–1,138] | 1.3 |
| Schizophrenia | 98.7 | 91.2% | $911 [809–1,26] | 1.4 |
| Substance use disorders | 1,847.5 | 92.4% | $3,325 [3,36–3,568] | 1.2 |
| *Diabetes and kidney diseases* | | | | |
| CKD | 1,456.1 | 97.6% | $14,376 [12,291–16,670] | 4.4 |
| Diabetes | 9,086.4 | 95.2% | $910 [790–1,44] | 2.0 |
| *Skin and subcutaneous diseases* | 21,662.3 | 86.2% | $303 [252–364] | 1.0 |
| *Sense organ diseases* | 6,909.4 | 87.7% | $1,020 [907–1,116] | 1.0 |
| *Musculoskeletal disorders* | | | | |
| Gout | 758.9 | 93.6% | $409 [334–498] | 2.0 |
| Low back and neck pain | 14,952.6 | 94.1% | $861 [747–987] | 1.1 |
| Osteoarthritis | 3,664.1 | 98.1% | $3,239 [2,714–3,848] | 1.7 |
| Rheumatoid arthritis | 306.8 | 95.7% | $424 [359–480] | 0.9 |
| *Other non-communicable diseases* | | | | |
| Congenital birth defects | 835.0 | 94.0% | $4,150 [3,748–4,577] | 1.9 |
| EMBI disorders | 11,813.9 | 93.2% | $2,386 [2,071–2,742] | 1.5 |
| Gynecological diseases | 15,033.3 | 83.2% | $2,318 [2,082–2,578] | 1.2 |
| Hemoglobinopathies and hemolytic anemias | 5,307.4 | 92.5% | $3,371 [2,862–3,803] | 1.7 |
| Oral disorders | 657.6 | 85.7% | $1,589 [1,384–1,819] | 1.2 |
| *Communicable diseases* | | | | |
| HIV/AIDS | 329.9 | 81.4% | $623 [543–694] | 1.2 |
| *Risk factors* | | | | |
| Hyperlipidemia | 13,465.3 | 96.4% | $204 [176–234] | 0.6 |
| Hypertension | 17,727.3 | 93.7% | $563 [501–635] | 1.1 |
| Obesity | 9,915.2 | 94.1% | $1,491 [1,346–1,613] | 1.0 |
| Tobacco use | 2,580.4 | 94.4% | $2,516 [2,267–2,810] | 1.0 |

*Top three excluding the disease itself and all the "other" categories (7 others).

CKD, chronic kidney disease; COPD, chronic obstructive pulmonary disease; EMBI disorders, endocrine, metabolic, blood, and immune disorders; IHD, ischemic heart disease.

When we consider the total spending associated with combinations (i.e., including the coefficients of the intercept, covariates, single conditions, and three dyads in the case of triads), the top five highest spending with observed prevalence greater than 50 per 100,000 were combinations of EMBI disorders + anemias + skin and subcutaneous diseases ($7,120 [6,899,7,319]), chronic kidney disease + EMBI disorders ($6,730 [6,387,7,29]), EMBI disorders + anemias + hypertension ($6,325 [5,762,6,950]), cirrhosis + EMBI disorders ($5,370 [5,236,5,528]), ischemic heart disease (IHD)-related conditions + hypertension + hyperlipidemia ($5,339 [5,115,5,574]). Note that while these have the highest estimated spending, they do not have the highest observed prevalence. Instead, when we further focus on combinations with at least 1% prevalence, we found the following combinations with the highest spending: IHD + hyperlipidemia + hypertension ($5,234 [4,538,5,826]), EMBI disorders + anemias ($4,961 [4,381,5,483]), gynecological diseases + anemias ($3,243 [3,126,3,350]), gastritis and duodenitis, peptic ulcer disease + obesity ($3,064 [2,978,3,158]), IHD + hyperlipidemia ($3,038 [2,759,3,295]).

**Table 2. Summary statistics of study population.**

| Characteristic | Statistic |
|---|---|
| Total | 16,288,894 |
| Sex | 56.2% Female |
| Age | 42.3 (SD 13.5) |
| Region | |
| Northeast | 19.0% |
| North Central | 20.5% |
| South | 44.2% |
| West | 16.2% |
| Unknown | 0.3% |
| Mean (median) annual spending | Mean 6,388.0<br>Median 633.3<br>SD 36,561.7 |
| Mean (median) number of chronic conditions | Mean 2.6<br>Median 2.0<br>SD 2.7 |

## Multimorbidity-adjusted spending for single chronic conditions

The majority of individuals with one of the 63 chronic conditions had at least one or more of the remaining 62 conditions. The highest proportions were recorded among individuals with stroke (98.7%), gastrointestinal gland cancers (98.6%), rheumatic heart disease (98.6%), tracheal, bronchus, and lung cancer (98.4%), and peripheral vascular disease (98.4%); the lowest proportions included HIV/AIDS (81.4%), gynecological diseases (83.2%), attention-deficit and hyperactivity disorder (84.9%), oral disorders (85.7%), and skin and subcutaneous diseases (86.2%). The average across all conditions was 93.9%.

With multimorbidity adjustment, the following 10 chronic conditions had the highest multimorbidity-adjusted spending per treated case: chronic kidney disease ($14,376 [12,291,16,670]), blood cancers ($9,387 [8,627,10,071]), gallbladder and biliary diseases ($7,509 [6,825,8,124]), cirrhosis ($6,465 [6,090,6,930]), brain and nervous system cancer ($6,298 [5,334,7,386]), IHD ($6,029 [5,529,6,529]), tracheal, bronchus, and lung cancer ($5,422 [4,874,6,085]), stroke ($5,395 [4,564,6,326]), atrial fibrillation and flutter ($5,147 [4,864,5,475]), and inflammatory bowel disease ($4,697 [4,594,4,813]). Among these, chronic kidney disease, cirrhosis, IHD, and inflammatory bowel disease had observed prevalence rates greater than 1,000 per 100,000 people (Fig 1).

Comparing the multimorbidity-adjusted spending estimates to non-adjusted spending, we found that 50 conditions (among 63) had higher spending after adjustments, 7 had less than 5% difference, and 6 had lower spending (Table 1). The top five conditions with the largest increase in spending after multimorbidity adjustment include chronic kidney disease (4.4 times increase), brain and nervous system cancer (3.7 times), atrial fibrillation and flutter (3.1), anxiety disorders (3.0), and rheumatic heart disease (2.9). The top five conditions with the largest decrease in spending after adjustment include breast cancer (0.6), hyperlipidemia (0.6), headache disorders (0.7), interstitial lung disease (0.8), and rheumatoid arthritis (0.9).

Decomposition of the multimorbidity-adjusted spending by the type of combination (single, dyads, and triads) for conditions with the highest spending per treated case is in Fig 2. The sizes of the contribution of dyads and triads are a function of the estimated spending associated with the disease combination as well as the observed prevalence of the combination among people with the condition. For example, for chronic kidney disease, less than 25% (in gray) is attributed to people having just chronic kidney disease, while approximately 50% (in

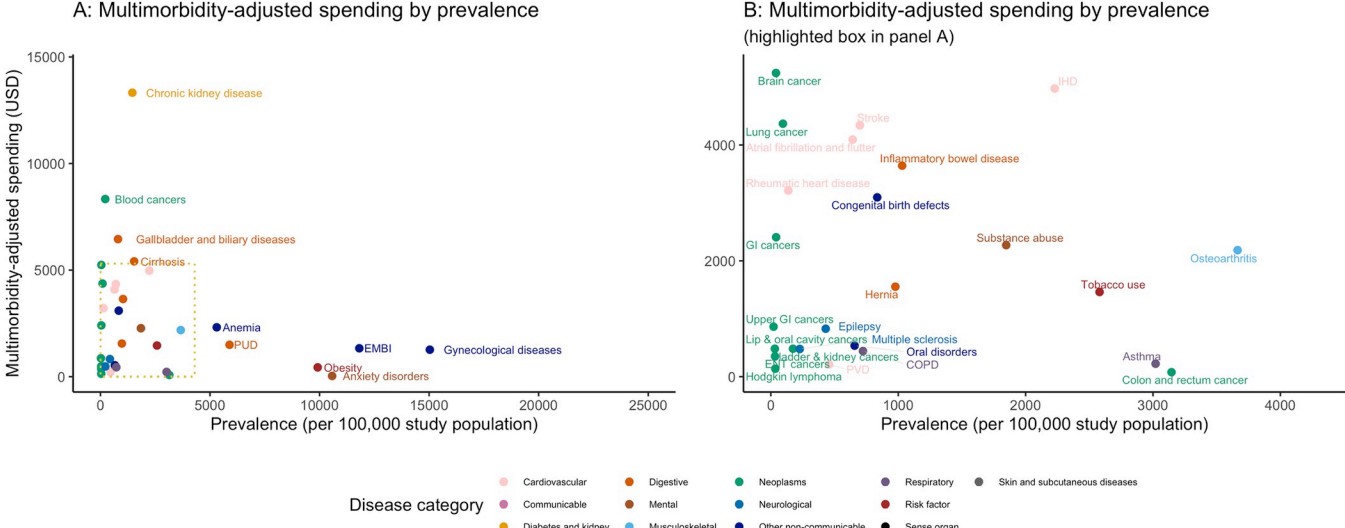

**Fig 1. Multimorbidity-adjusted spending per treated case and observed prevalence for 63 chronic conditions.** Abbreviations: ADHD, attention-deficit/hyperactivity disorder; COPD, chronic obstructive pulmonary disease; EMBI, endocrine, metabolic, blood, and immune disorders; ENT, ear, nose, throat; GERD, gastroesophageal reflux disease; GI, gastrointestinal; hernia, inguinal, femoral, and abdominal hernia; IHD, ischemic heart disease; PHD, peripheral vascular disease; PUD, peptic ulcer disease; sarcoidosis, interstitial lung disease and pulmonary sarcoidosis; sense, sense organ diseases; skin, skin and subcutaneous diseases.

yellow) is the increase due being in combination with a second condition, and the remaining 25% (in purple) is the increase due to being in a triad. Conditions such as chronic kidney disease and brain and nervous system cancer have higher contribution of spending from dyads and triads not only because spending on its combinations are high, but also because people with these conditions have higher probabilities of having multimorbidity (see Table 1, 97.6 and 95.9%, respectively) (Fig 2).

Finally, for conditions with the highest spending per treated case, we further decomposed the contribution of dyads and triads (the yellow and purple bars in Fig 2) by the type of coexisting disease categories (Fig 3). This graph shows how other condition contribute to the increase in spending for the index condition. For the purpose of comparison, we present stacked bar plots capped at 100%, but the actual spending estimates for each disease can be found in Table 1. First, spending associated with having only the index condition itself is presented in gray, ranging from less than 25% in chronic kidney disease to over 50% in inflammatory bowel disease and IHD, consistent with what was shown in Fig 2. Second, each color represents one major disease category, and the size of the bars represent the prevalence-weighted sum of the multimorbidity-adjusted spending associated with the coexisting disease category in combination with the index condition. For example, for chronic kidney disease (the first bar in Fig 3), we estimated overall multimorbidity-adjusted spending as approximately $14,300 (Table 1). In the figure, we see that the largest contributions to spending increase come from being in combination with "other non-communicable disease" (such as EMBI disorders), followed by cardiovascular and respiratory diseases, contributing to approximately 20%, 15%, and 10% of the total estimate, respectively. In other words, in the multimorbidity-adjusted estimates for chronic kidney disease, close to half (approximately $7,000) is due to being in combination with these disease categories. For the two cancers included in this figure, we see that other neoplasms account for the largest share of increases, and specifically for brain and nervous system cancer, we also see a large contribution from neurological disorders.

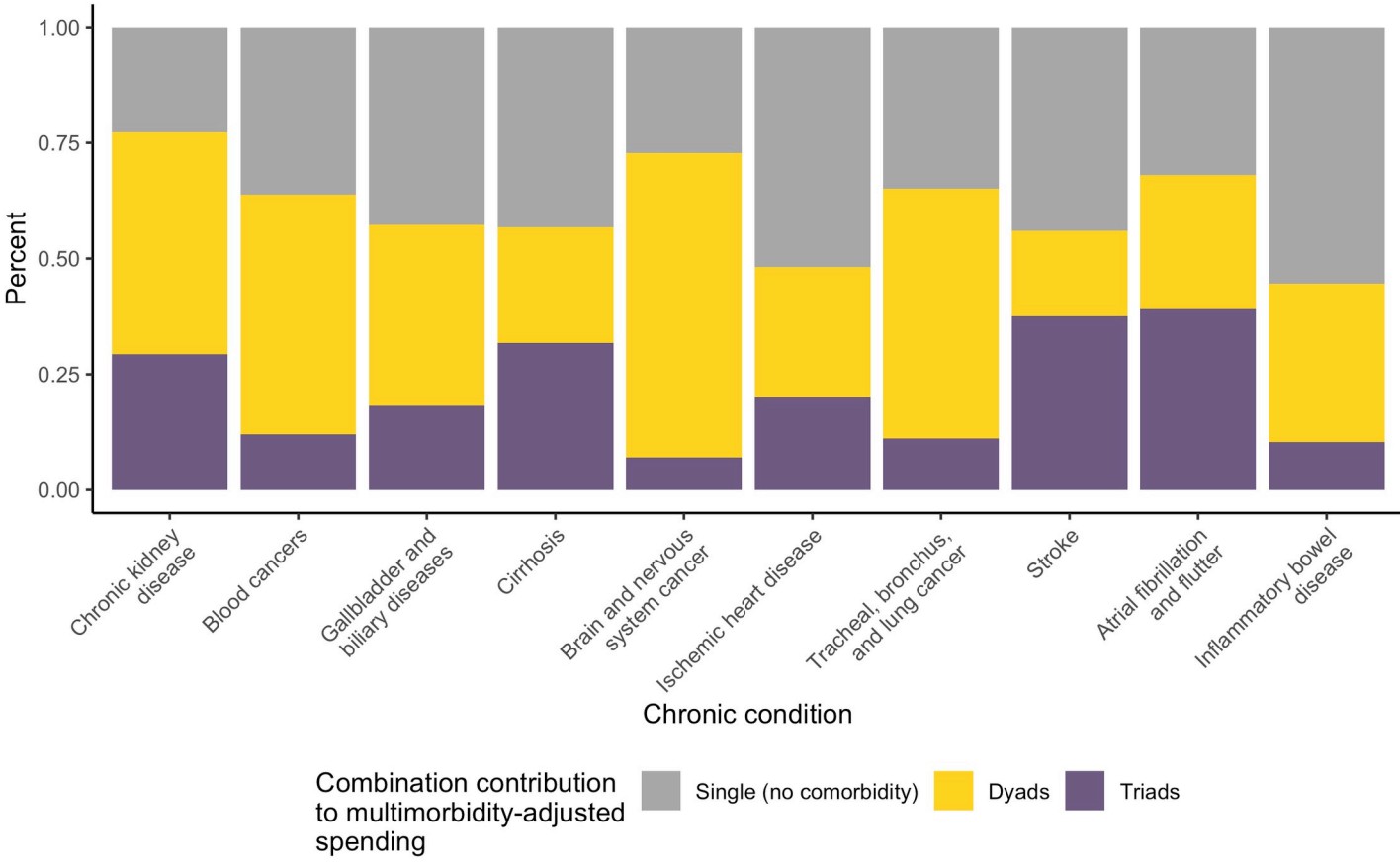

**Fig 2. Decomposition of multimorbidity-adjusted spending by the type of combination (single, dyads, triads) for chronic conditions with the top 10 highest spending per treated case.**

## Discussion

The relationship between multimorbidity and health spending has been listed as one of the research priorities proposed by the Academy of Medical Sciences [33]. This paper takes two perspectives—assessing spending on combinations of chronic conditions and assessing spending on a single chronic conditions with a multimorbidity adjustment—to provide different interpretation of a large dataset and more accurate spending estimates on a comprehensive list of conditions. Below, we highlight four key takeaways from this study.

First, across different sets of analyses, we consistently found chronic kidney disease, blood cancers, cirrhosis, and IHD to be associated with high spending per treated case, high observed prevalence, and contributing the most to spending when in combination with other chronic conditions. Preventing these conditions from occurring could mean large savings not only from its own treatment spending but also from its effect on spending on other conditions. While this study does not allow us to explain why, our results could guide further research designs, such as, for example, the distribution of spending by outpatient and inpatient services, and whether each have different additive/super-additive patterns. Reasons for super-additive spending could include greater utilization frequency, more complex disease trajectories due to disease and/or medication interactions, and lack of coordination between services [34]. Some have suggested that most clinical trials focus on treatments for single conditions and exclude participants with multimorbidity; therefore, even if the clinical guidelines are tailored for

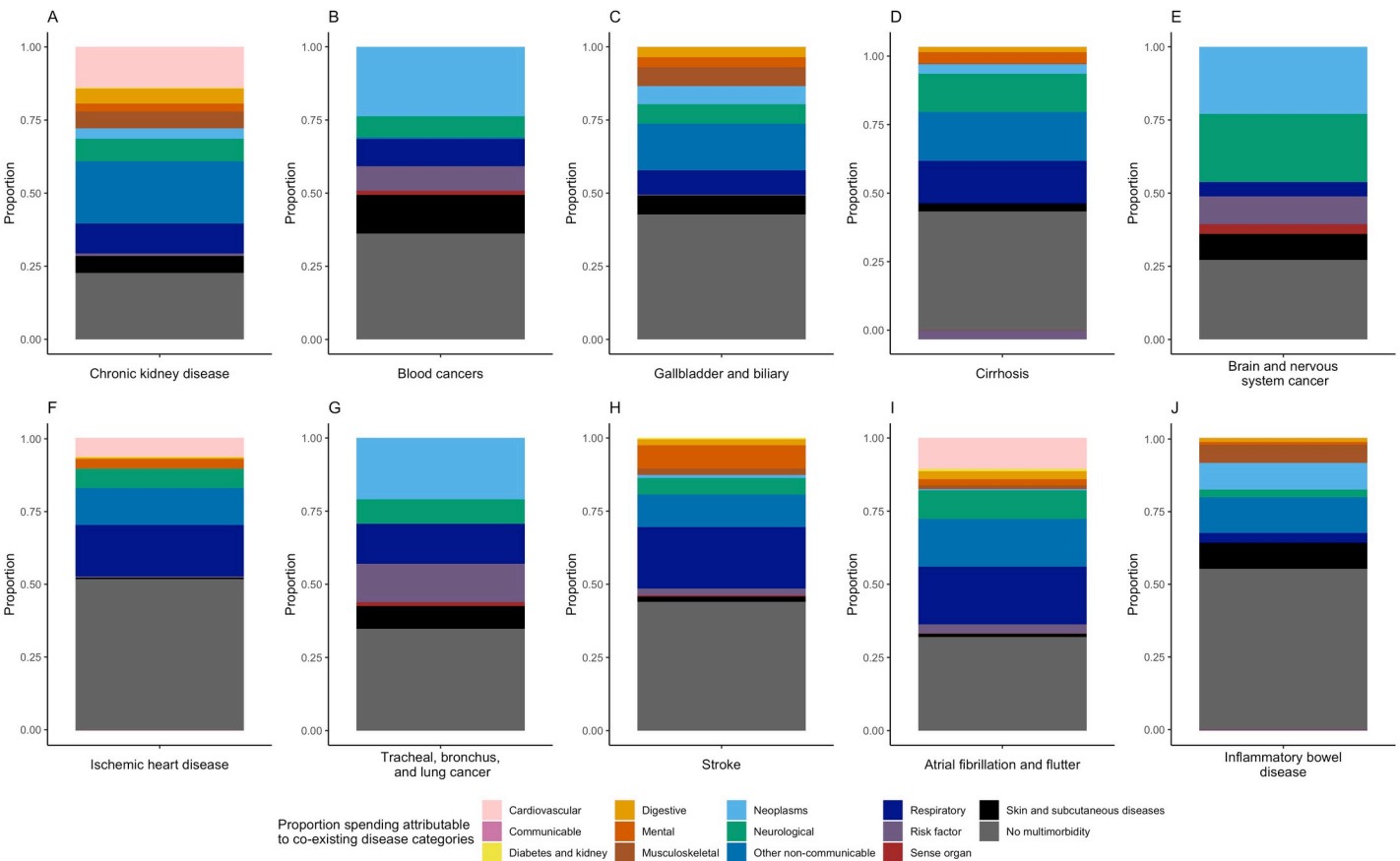

**Fig 3. Decomposition of multimorbidity-adjusted spending by multimorbidity disease categories, for chronic conditions with the top 10 highest spending per treated case.**

people with multimorbidity, the treatments may not be the most appropriate for this population [10]. Interventions such as care coordination, improvements in patient–provider communication, or targeting common risk factors have been proposed as means to reducing multimorbidity spending [34]. On the other hand, previous studies have pointed to how treatment for concordant conditions, defined as conditions similar in risk profile and management, could benefit from synergistic effects because services for chronic conditions that are treated by the same specialty (such as cardiology or internal medicine) may be more coordinated or are under more favorable payment schemes, leading to lower total spending than the spending for single conditions combined [35–39].

Second, contrary to common belief, multimorbidity does not always lead to greater spending than the sum of the spending associated with having these conditions separately: 40% of combinations did not have super-additive spending. In this study, the sub-additive spending was commonly found among people with hyperlipidemia or breast cancer alongside another chronic condition (i.e., conditions with lower spending post-adjustment; see Table 1). Sub-additive spending could mean that there may either be synergistic or harmful effects in how these patients are seeking or receiving care. Blakeley and colleagues hypothesized that sub-additive spending could also be due to down-prioritization of treatment for the comorbidities [17]. Compared to concordant diseases, discordant diseases, defined as those not directly related in pathogenesis or management, were found previously to either have zero or negative effects on one another [35–39].

Third, decomposition of spending associated with the type of combination (single/dyads/triads) and categories of multimorbidity (e.g., cardiovascular, neoplasms) allows us to more clearly identify and target common patterns of multimorbidity associated with high spending. For example, a large increase in spending associated with stroke, atrial fibrillation and flutter, and chronic kidney disease come from triads (Fig 2), which are likely more complicated, and a more concerted management effort to reduce the prevalence and/or spending associated with these single conditions could yield highly cost-effective results.

Fourth, a large proportion of our study population has multimorbidity, even among our study population who are under age 65. A previous study found that much of the recent growth in health spending in the Medicare population is due to increasing number of people with multimorbidity [40]. While this study focused on adults not eligible for Medicare, it is reasonable to assume that the increasing prevalence of multimorbidity in this population also is contributing to the substantial increase in spending. Consistent with previous studies, the highest observed prevalence rates were found among combinations of hyperlipidemia, hypertension, diabetes, skin and subcutaneous diseases, gynecological diseases, low back and neck pain, and EMBI disorders [2,12,18].

It is difficult to benchmark our results because no other study had a similar scope. We chose five studies that are quite different but relevant for triangulation. Dieleman and colleagues estimated population-level spending for a comprehensive set of conditions, and among chronic conditions in all ages, including ages 15 to 64, they found the largest positive increases in chronic kidney diseases, alcohol disorders, diabetes mellitus, chronic obstructive pulmonary disease (COPD), and skin and subcutaneous diseases, and largest decreases in atrial fibrillation and flutter, urinary diseases, gynecological diseases, bipolar disorders, and depressive disorders [22]. While not the main focus of this study, among these conditions, we observed the same directional changes in only half of those listed above. Possible explanations include the difference in estimating spending for a population (instead of per treated case in our study), the assumption of assessing multiplicative (instead of additive) effects, its focus on inpatient and nursing facility spending (instead of inpatient and outpatient), and the inclusion of more conditions and combinations. Second, DEX, which looks at spending at the population level for all ages and types of payers, also identified IHD, hypertension, and urinary diseases among conditions with the highest spending [1]. Though, note that the reason for high population-level spending could either be because of high individual-level spending, high prevalence, or a mix of both (e.g., medium-level spending times medium-level prevalence could lead to relatively high overall spending). Third, Tran and colleagues performed a meta-analysis based on 15 studies for 11 most frequently reported dyads, and estimated mean costs to be between $13,270 (hypertension + musculoskeletal disorder) and $85,820 (cancer first year after diagnosis + mental health conditions) (2021 International Dollars). These numbers are larger than our estimates; however, they are not comparable due to differences in study population, age, study design, among other factors. Among the studies reported by the review and used similar data sources, we found qualitatively similar results reported for rheumatoid arthritis, chronic kidney disease, and diabetes [41–43]. Fourth, compared to Rezaee and Pollock who estimated total outpatient spending for conditions in a large US health system between 2008 and 2013, we reached a similar conclusion that hyperlipidemia, hypertension, and combinations of these two conditions along with other conditions are prevalent and costly [12]. They estimated that one additional chronic condition was associated with increased spending of approximately $600 but did not report further on the types of combinations nor did they distinguish between super- or sub-additive spending [12]. Finally, the study by Blakely and colleagues estimates individual-level spending estimates from New Zealand [17]. Using higher-level disease categories, they found the highest single-condition spending in chronic

lung, liver, and kidney diseases, and the highest spending in combinations of cancer + neurological disorders, cardiovascular diseases + chronic lung, liver, and kidney diseases.

This study has several strengths and limitations. One key strength is the application of a large and comprehensive set of 63 chronic conditions (which are composed of 166 most-detailed GBD causes), overcoming limitations faced by existing studies due to lack of data availability or computational limitations. Moreover, we were able to study combinations of up to three conditions per individual, not limited to a small set or combinations of only two conditions [17,44]. Second, this study is based on a comprehensive administrative claims database that encompasses a large population of over 16 million adults in the US, providing sufficient information on the diagnoses and spending of enrollees. Compared to other datasets such as self-reported data, administrative data are often more reliable and allows for easier comparisons across studies [45]. However, the data do not include information on functional limitations and disease duration, which could provide more insights into the relationship between conditions and spending [46]. Due to the structure of administrative claims data, this paper takes a healthcare payer's perspective, which likely underestimate the true cost of multimorbidity because we do not account for out-of-pocket payments or indirect costs. Third, we analyzed the relationship between multimorbidity and spending from two perspectives—single and combination of conditions. Instead of classifying combinations into one as primary diagnosis and the others as comorbidities as done in other studies, we took a multimorbidity perspective and distributed the spending across all conditions based on weights provided by a set of separate regression models [21,22].

The limitations of this study include the following. First, MarketScan data are a convenience sample—it is not representative of the commercially insured US adult population. For example, MarketScan data draw disproportionately from the southern region of the US [47]. We lack data on income but we assume this population has higher income than the general population since this is an employment-based claims database. We therefore cannot conclude that our results are generalizable at the national level. Similarly, this study focused on adults younger than 65 and the results thus should not be generalized to the Medicare population, who have the highest prevalence of chronic conditions. This is also a cross-sectional analysis focusing on 2018, with a gap of four years from the time of this writing in 2022. Ideally, we would study multiple years to identify time of diagnose as well as minimize the impact of random variations in health spending between years, but due to data access and financial restrictions at the time of writing, we are unable to do so. This has been found as a common limitation across studies on this topic [8]. The COVID-19 pandemic also significantly delayed our ability to analyze data. We also only have data on spending associated with a diagnosis, but we do not know about disease severity beyond what is represented in the ICD codes. Second, our spending variable only considers expenses incurred during inpatient or outpatient visits and does not include spending on pharmaceuticals or medical products incurred outside of these visits, nursing facility spending, nor indirect costs such as opportunity costs, transportation costs, and costs due to lost productivity—which are likely substantially higher among people with certain combinations than others. While the dataset has information on pharmaceutical spending for enrollees outside of visits (such as retail settings), we did not include it because of the difficulty in mapping pharmaceuticals to exact diagnoses (since drugs may be prescribed for multiple purposes). It is possible that individuals may have chosen pharmaceutical products over seeking provider services, which would lead to an underestimation of inpatient/outpatient spending for certain diseases. Third, the model structure implicitly assumes that the contributions of additional conditions on spending are additive and not multiplicative [46]. Related, the study design only allows for non-causal interpretation of the results. Fourth, while our list of chronic conditions, as well as the approach of diagnosing

chronic conditions, follow the approach set by the CMS [2,43], there may be a more precise definition of categorizing diseases as chronic that are more suitable. We also do not include injuries and non-chronic conditions, some of which might have large impacts on spending. Fifth, due to computational limitations, we were only able to estimate spending for dyads and triads, though we speculate that combinations beyond three conditions do not contribute much to the multimorbidity adjustment.

This paper offers several insights into how the economic and health burden of multimorbidity could be better understood and provides a systematic method for measuring spending on multiple chronic health conditions that could be replicated elsewhere. In the midst of a surging health spending globally, and especially in the US, pinpointing high-prevalence, high-spending conditions and disease combinations, as especially conditions that are associated with larger super-additive spending, could help policymakers, insurers, and providers prioritize and design cost-effective interventions to improve treatment effectiveness and reduce spending.

## Supporting information

**S1 RECORD Checklist. RECORD checklist.**
(DOCX)

**S1 Appendix. Technical appendix.** Table A1. List of most-detailed GBD causes determined as chronic conditions. Table A2. Chronic conditions and its most-detailed GBD causes. Table A3. Summary statistics of 50 SGD model runs. Table A4. Summary statistics of study population. Table A5. Covariates and their regression coefficients. Table A6. Observed prevalence and estimated multimorbidity-adjusted annual spending per treated case for the "other" conditions. Table A7. Regression coefficients for disease combinations with prevalence rate of greater than 100 per 100,000 (ordered by spending).
(DOCX)

## Author Contributions

**Conceptualization:** Angela Y. Chang, Joseph L. Dieleman.

**Data curation:** Angela Y. Chang, Dana Bryazka, Joseph L. Dieleman.

**Formal analysis:** Angela Y. Chang, Dana Bryazka, Joseph L. Dieleman.

**Investigation:** Angela Y. Chang.

**Methodology:** Angela Y. Chang, Joseph L. Dieleman.

**Validation:** Angela Y. Chang, Joseph L. Dieleman.

**Visualization:** Angela Y. Chang, Dana Bryazka, Joseph L. Dieleman.

**Writing – original draft:** Angela Y. Chang.

**Writing – review & editing:** Angela Y. Chang, Dana Bryazka, Joseph L. Dieleman.

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
