## [Editor Report · Decision Letter 0]

19 Aug 2022

Dear Dr Chang, 

Thank you for submitting your manuscript entitled "Estimating health spending associated with chronic multimorbidity in 2018: an observational study among adults in the United States" for consideration by PLOS Medicine.

Your manuscript has now been evaluated by the PLOS Medicine editorial staff as well as by an academic editor with relevant expertise and I am writing to let you know that we would like to send your submission out for external peer review.

Please re-submit your manuscript within two working days, i.e. by Aug 23 2022 11:59PM.

Kind regards,

Dr. Philippa Dodd, MBBS MRCP PhD

Senior Editor

PLOS Medicine

---

## [Decision Letter · Decision Letter 1]

13 Dec 2022

Dear Dr. Chang,

Thank you very much for submitting your manuscript "Estimating health spending associated with chronic multimorbidity in 2018: an observational study among adults in the United States" (PMEDICINE-D-22-02788R1) for consideration at PLOS Medicine. 

[LINK]

In light of these reviews, I am afraid that we will not be able to accept the manuscript for publication in the journal in its current form, but we would like to consider a revised version that addresses the reviewers' and editors' comments. Obviously we cannot make any decision about publication until we have seen the revised manuscript and your response, and we plan to seek re-review by one or more of the reviewers. 

We expect to receive your revised manuscript by Jan 03 2023 11:59PM. Please email us (plosmedicine@plos.org) if you have any questions or concerns.

We look forward to receiving your revised manuscript. 

Sincerely,

Philippa Dodd, MBBS MRCP PhD

PLOS Medicine

plosmedicine.org

GENERAL

Please respond to all editor and reviewer comments detailed below, in full.

Please note reviewer comments below requiring the use of language which clearly clarifies your descriptions and definitions/categorizations 

Please address the conceptualization issues identified

Thank you for reporting (we think!) according to RECORD – see below, please clarify. 

Please review the guidance using the link below regarding reporting of studies: 

http://www.equator-network.org/?post_type=eq_guidelines&eq_guidelines_study_design=economic-evaluations&eq_guidelines_clinical_specialty=0&eq_guidelines_report_section=0&s=

We suggest you report in line with RECORD (CHEERS is targeted to economic evaluations of interventions and STROBE is epidemiological reporting). Please provide the relevant completed checklist. In the checklist, please include sufficient text excerpted from the manuscript to explain how you accomplished all applicable items.

Please include the relevant associated checklist and indicate in your statement in the methods section where it can be located

DATA AVAILABLIITY STATEMENT

The Data Availability Statement (DAS) requires revision. For each data source used in your study: 

ABSTRACT

Please structure your abstract using the PLOS Medicine headings (Background, Methods and Findings, Conclusions).

Please combine the Methods and Findings sections into one section, “Methods and findings”.

Abstract Background: Provide the context of why the study is important. The final sentence should clearly state the study question.

Abstract Methods and Findings:

Please ensure that all numbers presented in the abstract are present and identical to numbers presented in the main manuscript text.

Please include the years during which the study took place, length of follow up, and clearly define the main outcome measures.

Please include the actual amounts and/or absolute risk(s) of relevant outcomes (including NNT or NNH where appropriate), not just relative risks or correlation coefficients. (Example for absolute risks: PMID: 28399126). 

Please define the numerical values contained within square parentheses (UIs as per your main methods/results).

Please also quantify the main results with p values (as well as UIs) 

Please include any important dependent variables that are adjusted for in the analyses.

In the last sentence of the Abstract Methods and Findings section, please describe the main limitation(s) of the study's methodology.

Abstract Conclusions:

Please address the study implications without overreaching what can be concluded from the data; the phrase "In this study, we observed ..." may be useful.

Please interpret the study based on the results presented in the abstract, emphasizing what is new without overstating your conclusions.

Please avoid vague statements such as "these results have major implications for policy/clinical care". Mention only specific implications substantiated by the results.

Please avoid assertions of primacy ("We report for the first time....")

AUTHOR SUMMARY

Thank you for including an author summary

Line 79: “Few studies have explored whether the combinations….” suggest “whether different combinations of conditions…” or something similar

Line 83: “16+ million…” suggest over 16 million

Line 95: “Multimorbidity adjustments should be done…” suggest performed in place of done

Please remove the data availability statement form the end of the author summary and include only in the manuscript submission form

INTRODUCTION

If there has been a systematic review of the evidence related to your study (or you have conducted one), please refer to and reference that review and indicate whether it supports the need for your study.

METHODS

Line 167: “…Strengthening the Reporting of Observational Studies in 168 Epidemiology (RECORD) guideline.” Should read, “REporting of studies Conducted using Observational Routinely collected health Data (RECORD) guideline.”

Please remove role of the funding source from the end of the methods section and include only in the manuscript submission form

FIGURES and TABLES

To improve accessibility to those with color blindness, please consider avoiding the use of green and/or red

Please provide an appropriate caption/legend for all figures and tables such that its contents are clearly described without the need to refer to the text.

Please provide a table showing the baseline characteristics of the study population

Table 1 (and throughout) where you report adjusted analyses, please also provide the unadjusted analyses for comparison, and please indicate in an appropriate caption or footnote which factors are adjusted for

Figures 1 and 2: are not very accessible to the reader – text overlies other text and makes it unreadable. The legend refers to sub-categories which the specific conditions presented in the graph fall into, this can be confusing to the reader. Please revise accordingly

Figure 3: in isolation, it is difficult to interpret what this figure shows. Is combination 1, 2, 3 (as referred to in the legend) the equivalent of single, dyads and triads (as referred to in the title)? Please revise to improve accessibility in-line with previous comments

Figure 4: please see reviewer comments which we agree with, and revise accordingly

DISCUSSION

Please remove the sub-heading conclusion such that the discussion reads as a continuous piece of prose

Please remove statements at line 602 and 603 and include only in the masncuript submission form

SUPPORTING INFORMATION

Please provide appropriate captions for all tables

Please provide p-values where 95% UIs are reported

Please provide unadjusted analyses where relevant and in the caption/footnote indicate which factors are adjusted for

Table A1 and A2: Please define GBD 

Comments from the reviewers:

Reviewer #1: I enjoyed reading this paper, which is well-written and clearly organised. The study aims to (1) assess if disease combinations are super-additive, additive, or sub-additive in regard to health spending, and (2) estimate disease-specific health spending that accounts for multimorbidity. The first research aim is particularly relevant to policy and clinical practice as it may inform prioritization of disease combinations most likely to result in the highest healthcare spending. Using private insurance claims of 16m enrollees, the paper finds that 60% of them (pairs or triads) are super-additive. The chronic conditions with the highest multimorbidity-adjusted spending include CKD, cirrhosis, IHD, and inflammatory bowel disease. The authors claim that prioritizing interventions to reduce the prevalence and/or spending of these conditions could yield to cost-effective results. Below I provide some comments that the authors may consider to potentially improve the paper:

1. Conceptualisation of the health expenditure model: 

* The paper offers no rationale for the specification of the health spending equation (equation 1, page 7). How did the authors choose the independent variables? Some sort of conceptual framework, specific to multimorbidity such as the one by Zulman et al 2014 on multimorbidity interrelatedness, could have guided this piece. How do the authors think that important variables such as disease severity, time since diagnosis, or total number of conditions may be confounding existing results? 

* Why did the authors limit themselves to disease dyads and triads? Based on the characteristics of the study population this may be particularly relevant (94% of those with multimorbidity had more than three conditions). Did the authors at some point consider the possibility of characterising the actual disease combinations in the sample using cluster analysis?

2. Estimation of the health expenditure model:

* Which estimator was used in the stochastic gradient descent approach? How does this estimation approach account for the skewed (non-normal) distribution of health expenditures?

* How was goodness of fit of the model assessed? What percentage of the variability in individual health expenditures is this model able to explain?

3. Limitations of the sample used:

* The authors used a cross-sectional sample. Without adjusting for time since diagnosis or disease severity, the disease stage that the estimated expenditures pertain to is unclear. 

* How is the study sample different from a US nationally representative sample? On page 5, the authors state that differences have not been assessed before but I wonder if they could be at least approximated by comparisons to publicly available sources. How could the disease prevalence of a subsample of commercially insured enrolled differ from a nationally representative sample? I think this is an important point to understand the usability and generalisability of study findings. 

4. Consideration on study definitions:

* What is the rationale behind not restricting claims to primary diagnosis to generate a more accurate estimate of the cost of a chronic condition? (page 6, lines 205-206). Does that mean that under the cost of CKD, costs of acute events are also included?

5. Other conceptualisation issues:

* In my opinion, how generating multimorbidity-adjusted health expenditure estimates for *individual* chronic conditions helps understanding the economic burden of multimorbidity could be more clearly articulated. The paper seems focused on multimorbidity, yet its conclusions apply to individual conditions. How do the conclusions of this paper then relate to the design of interventions to improve health outcomes of individuals with multimorbidity? 

* The authors only considered costs but not health outcomes, so the conclusion on cost-effectiveness towards the end is hard to follow and unclear how it can be drawn from the study results. 

Other comments:

* From what perspective was annual spending computed? It seems to include provider payments but exclude out-of-pocket expenditures incurred by the beneficiary? What is the rationale behind this choice and how does that affect your disease-specific expenditure estimates?

* How could spending data be negative (line 184, page 6)?

Reviewer #2: Reviewer's comments

1. Summary of the research and general comment:

This is an interesting and topical multimorbidity study, which estimated the costs of disease combinations and explored the interaction of co-existing conditions within an individual and their impact on costs. The study is very well structured, with a large sample size and number of disease combinations. 

The main limitations are that the study did not capture cost components beyond outpatient and inpatient services and unable to account for changes through time; however these are well acknowledged in the limitation. The methods section relating to the decomposition of the change in multimorbidity-adjusted spending can be more elaborative.

In general, writing up a paper on the topic of (the cost of) multimorbidity is challenging. It is easy for readers to get lost and confuse themselves amidst the myriads of jargons, the overwhelming number of ways diseases are combined, and all the different ways of expressing the same term (e.g. when referring to the cost of a condition, it is important to emphasize whether the author is referring to a condition as part of a disease combination or a single condition by itself; etc.)

In light of such complexities, phrasing should be as accurate and consistent as possible and details on methodology, terminologies should be elaborated as much as possible, to enable readers (who are not experts on multimorbidity) to easily follow. It is commendable that the author gave various examples of specific conditions/dyads throughout the paper. It would be helpful to be a bit more elaborative in places (see points below), and to be consistent with the choice of terms. 

2. Discussion of specific areas for improvement:

A. Major comments:

1. Line 190-192: Can the author explain why injuries were excluded from the list of chronic conditions? For example, hip fracture rates in the US are amongst the highest in the world and links to long-term disability outcomes, similar to stroke. 

2. Line 301-305, the author described the third and fourth step in estimating spending associated with each individual health condition, adjusting for multimorbidity. It will be beneficial to the readers if the author explains the reasons for undertaking these steps. Perhaps, the author can also demonstrate this using an equation. 

3. Can the author elaborate more in the method section what they specifically did to arrive at the estimates for the total spending associated with combinations?

4. Did the author log-transform cost, or how did the author address non-linearity/non-normality/unequal variance in the data?

5. Line 420-427: "Decomposition of the change in multimorbidity-adjusted spending by the type of combination (single, dyads, and triads) for conditions with the highest spending per treated case". To the reviewer's understanding, this is the spending on single conditions, when taking into account other co-morbidities. For example in the case of CKD, is the below the correct interpretation of this finding?

* For MM-adjusted spending for CKD, less than 25% is attributed to the base cost itself (in the case of an individual with only 1 condition, i.e. CKD). The yellow stripe (which represents around 50% of the total), is the increase in cost of CKD due to the interaction with the second condition - in the case of dyad. The red (which represents over 25% of the total), is the increase in cost of CKD due to the interaction with a third condition - in the case of triad.

* Does it imply that individuals with only 1 condition (CKD only) will incur lower cost than the adjusted estimate above and proportionate to the green stripe in the graph?

* Is this the increase 'on average'? For example, if CKD is comorbid with 2 other conditions (triad), the increase in CKD cost is the same on average and not particular to any specific conditions?

The reviewer advises that the author describes more in this section, to help readers understand what is being explored. If the above bullet points are the correct way to interpret this result, perhaps the author can consider elaborating it in such a way - citing the specific breakdowns (25%, 50%,…) as an example.

6. Line 429: "Comparing pre- and post-multimorbidity adjustment spending estimates, we found that 50 conditions (among 63) had higher spending after adjustments, 7 had less than 5% difference, and 6 had lower spending (Table 1)."

The author mentions pre-adjusted costs but has not presented these anywhere, or how it was calculated. The reviewer deems this necessary, especially for the latter (to be included in the method section). 

7. Line 442: "In other words, among these diseases, estimated multimorbidity-adjusted spending are at least twice higher than the non-adjusted spending estimates." 

- By "non-adjusted", is the author referring to the average cost of this specific condition among those with only 1 condition (base cost)? 

- By "twice higher", does the author mean "double"? The reviewer suggests that the author rephrases "twice higher" to "double" for more accuracy and clarity. 

- The author may also consider revising Line 442-443 as follow, so as to clearly distinguish between the spending for individual conditions and the spending for combinations. 

"In other words, the estimated multimorbidity-adjusted spending for these individual diseases are at least, double that of their respective non-adjusted estimates." 

8. Similarly, for Line 443: "For chronic kidney disease, we observe an increase of more than three times", while the reviewer understands the point, it may be a bit ambiguous to others. The author may consider revising this sentence to: 

"For chronic kidney disease, spending increased fourfold after MM-adjustment." 

(This is a safe way to phrase and unlikely to be misinterpreted.)

9. Line 438-448:

For this section, the reviewer would also like to clarify for enhanced understanding:

* In the example of CKD, the different colors show the increase in cost for CKD when CKD is comorbid with other conditions. For example, when comorbid with "other NCD" (bright green), the cost of CKD doubles that of the base cost (the base cost is the cost of CKD in the case of an individual with only 1 condition, i.e. CKD). Is this the correct way to understand?

* Is the change in CKD cost different when it is comorbid with 2 other conditions simultaneously instead of 1 (in the case of triads)? For example, when CKD is comorbid with both CVD and 'other NCD' at the same time, is the increase in cost of CKD simply additive of the thickness of their respective stripes?

10. Line 446: "For cancers, we see that other cancers account for the largest share of increases, and specifically for brain and nervous system cancer we also see a large contribution from neurological disorders." 

- By 'other cancers', the reviewer assumes the author is referring to 'neoplasms' in Figure 3, however 'neoplasms' may be benign (not cancer) or malignant (cancer). The author should refrain from using the term 'other cancers'.

11. It would be useful if the author also presented in the Appendix the estimated spending for all disease combinations with a prevalence rate greater than 100 per 100,000 (or a higher threshold, in case the number of combinations is too high).

12. The estimated spending for disease combinations reported in the paper are on average much lower than those reported in other studies from the US (see 10.1186/s12916-022-02427-9), some of which also analyzed MarketScan data. Does the author have more thoughts on why that is?

13. Exploring the distribution of cost components (i.e., outpatient vs inpatient) of costly disease combinations would also be useful in identifying the driver of high costs. This is particularly relevant in the discussion, where the author discusses the potential reasons for super-additive spending. 

14. The strengthening of the integrated primary care system is an important intervention to target as the burden of multimorbidity increases. It would be relevant to mention this alongside prioritizing prevention, in the Author Summary (Line 94-99). The author has mentioned service coordination, patient-provider communication in the discussion, and these are aspects of integrated care. In addition, provider-provider communication and self-management are important aspects to also consider. Provider financing models (fee-for-service vs capitation) may also affect the level of service utilization/cost. 

15. Line 481-483: Here it would be useful to give an example of triads or dyads that had sub-additive spending.

B. Minor comments:

1. Line 167-168: Please cite the RECORD guideline.

2. Line 198: Please remove "to".

3. The "==" in equation 3 should just be "=".

4. Line 377-380: Are the reported costs in brackets total or incremental? If they are indeed incremental (i.e. the overall increase in cost due to the interaction of the component diseases), then perhaps the author should clearly specify that, and also add a "+" before each Dollar sign. 

E.g. blood cancers + hemoglobinopathies and hemolytic anemias (henceforth anemias) (+$3227, 95%UI [2541-3905]).

5. Line 382: Please add '$' to "cirrhosis + hyperlipidemia (-610 [-838- -370])" for consistency.

6. Line 384: Please revise '$-' to '-$' in "hyperlipidemia ($-545 [-727- -363])" for consistency.

7. Line 422: Consider changing "size" to "sizes".

8. Line 473: Please revise "point to" to "have pointed to".

9. The reviewer finds the style of visualization in Figure 1 a bit difficult to follow.

10. For Figure 4, please name the ten graphs from A to J for ease of referencing.

11. Figure 4:

* Each graph gives the false impression that it represents the proportions of cost of various conditions co-existing in the same individual. Readers may have to refer to the main text to understand. In theory, graphs and charts should be self-explanatory; readers should be able to infer from them immediately without having to read the main text. If this is the most optimal way to visualize the result, the author may want to be more descriptive in the main text (as suggested in the points made above) as well as add a short sentence on what is going on in the graphs/what was done to get there (also for Figure 3).

* The bright red and/or dark orange stripe are often small and shadowed by the dark red bar (due to the color palette, and as these stripes tend to be very thin), the author may consider changing the bright red/dark orange to other distinctive colors or change the orders of the colors so that distinctive colors are next to each other. 

* In the legend, dark orange refers to both diabetes and CKD merged together. In the first graph for CKD, is it correct that there is a thin dark orange stripe that represents both Diabetes and CKD? If yes, would there be an overlap here? If that stripe is actually not dark orange, but a bright red stripe (these two colors cannot be distinguished by the reviewer, as it is thin and shadowed by the dark red color of CVD), then is Diabetes missing from this graph? Where in fact, does the Diabetes and CKD stripe (dark orange) appear across the 10 graphs?

Reviewer #3: This study explores the relationship between multimorbidity and spending on inpatient and outpatient care among adults in the US.

Comments:

"This study is reported following the Strengthening the Reporting of Observational Studies in 168 Epidemiology (RECORD) guideline."

Can the authors please clarify if they followed STROBE or RECORD guidelines here?

Can the authors also please supply the associated checklist in the supplementary material?

"This study took a person-based regression approach - regressing a person's total 2018 health spending on health conditions indicators - in estimating spending per treated case [28,29]. We applied the following linear regression model:... "

and

"we applied a regression framework using the stochastic gradient descent (SGD) approach... we applied a lasso penalized regression model to shrink the coefficient values of these covariates"

and

"To ensure stability of the model results, we conducted 50 SGD model runs and bootstrapped the results across runs for 10,000 times to get the estimates for all coefficients"

The authors have applied technically appropriate methods, which they describe clearly within the article.

Similarly, the authors have conducted rigorous and comprehensive models for "Estimating spending associated with each individual health condition, adjusting for multimorbidity".

"First, to generate 95% uncertainty interval (UI) for spending associated with disease combinations, we bootstrapped the means from all the model runs for 10,000 times. Second, to generate UI for spending associated with single conditions, we ran Monte Carlo simulations (n=1,000 draws) while varying the estimates associated with the combination and the proportion of combination attributed to each single condition." 

The authors have conducted suitable analyses to help demonstrate the uncertainty and robustness of the study findings.

Overall, the authors have communicated the study Results accurately and the main study limitations have been suitably addressed in the Discussion.

Furthermore, the authors provide satisfactorily detailed methods and results in the supplementary material.

[LINK]

---

## [Decision Letter · Decision Letter 2]

9 Feb 2023

Dear Dr. Chang,

Thank you very much for re-submitting your manuscript "Estimating health spending associated with chronic multimorbidity in 2018: an observational study among adults in the United States" (PMEDICINE-D-22-02788R2) for review by PLOS Medicine.

I have discussed the paper with my colleagues and the academic editor and it was also seen again by 2 reviewers. I am pleased to say that provided the remaining editorial and production issues are dealt with we are planning to accept the paper for publication in the journal.

[LINK]

We look forward to receiving the revised manuscript by Feb 16 2023 11:59PM.   

Sincerely,

Philippa Dodd, MBBS MRCP PhD

PLOS Medicine

plosmedicine.org

Requests from Editors:

GENERAL

Thank you for your considerate and detailed responses to previous editor and reviewer requests. Please see below for further revisions which we require you address in full.

ABSTRACT

Line 72: please remove this statement and include only in the manuscript submission form

AUTHOR SUMMARY

Line 79: we would advise against the use of the word “anecdotally” perhaps, “Many would agree…” or something similar. Please revise. In addition, perhaps “…directed towards…” instead of “…focused on…”

Line 86: Please revise this statement for clarity and improved accessibility to the reader

INTRODUCTION

Line 156: “…to date, no study…” suggest the addition of “to our knowledge” as claims of supremacy can be risky

Line 162: “For example, we would have a more accurate spending on stroke because we would have not only the stroke-specific spending but also the additional or reduced amount of spending stroke incurs when in combination with other conditions.” Suggest moving this statement to an appropriate part of the methods section as it justifies/explains the benefit of the adjustment, as we understand things.

STATISTICAL REPORTING

Line 390: “…anemias (+$3111 [2679-3535])…” and line 393: “…hyperlipidemia (-$733 [-851- -620])…” I note the use of hyphens as well as the reporting of negative values which could be confusing to the reader. Suggest the use of commas instead

REFERENCES

Please ensure that in-text reference callouts are placed within square parentheses for example, line 112 should read as follows. “…two decades [1].

In your bibliography, please ensure that up to, but no more than, 6 authors are listed followed by et al, in cases where more than 6 authors contribute to a study.

Please ensure that journal name abbreviations used in the bibliography are those found in the National Center for Biotechnology Information (NCBI) databases

Please see our website for other reference guidelines https://journals.plos.org/plosmedicine/s/submission-guidelines#loc-references

SOCIAL MEDIA

To help us extend the reach of your research, please provide any Twitter handle(s) that would be appropriate to tag, including your own, your coauthors’, your institution, funder, or lab. Please detail any handles you wish to be included when we tweet this paper in the manuscript submission form when you re-submit the manuscript.

Comments from Reviewers:

Reviewer #2: The author has acknowledged, addressed and/or responded well to issues highlighted by the reviewer. The reviewer has no further comments/suggestions and congratulates the authors on this important study. 

Reviewer #3: Many thanks to the authors for responding to each comment in turn.

[LINK]

---

## [Editor Report · Decision Letter 3]

20 Feb 2023

Dear Dr Chang, 

On behalf of my colleagues and the Academic Editor, Professor Aaron Kesselheim, I am pleased to inform you that we have agreed to publish your manuscript "Estimating health spending associated with chronic multimorbidity in 2018: an observational study among adults in the United States" (PMEDICINE-D-22-02788R3) in PLOS Medicine.

Before we can publish your manuscript, we require that you make the following amendment: 

* Line 403 "($4961 [4381-5483])" please replace the hyphen with a comma.

PRESS

Best wishes,

Pippa 

Philippa Dodd, MBBS MRCP PhD 

PLOS Medicine